# Real-World Longitudinal Experience of Botulinum Toxin Therapy for Parkinson and Essential Tremor

**DOI:** 10.3390/toxins14080557

**Published:** 2022-08-17

**Authors:** Olivia Samotus, Yekta Mahdi, Mandar Jog

**Affiliations:** 1Department of Clinical Neurological Sciences, London Health Sciences Centre—Lawson Health Research Institute, 339 Windermere Road, A10-026, London, ON N6A 5A5, Canada; 2Schulich School of Medicine and Dentistry, University of Western, 1151 Richmond Street, London, ON N6A 3K7, Canada

**Keywords:** Botulinum toxin, upper-limb tremor, treatment, essential tremor, Parkinson’s disease

## Abstract

Background: Botulinum toxin type A (BoNT-A) therapy for upper-limb tremor has emerged as a promising option. However, it is unclear in real-world practices whether a technology-guided approach can compare with expert clinical assessments (including surface anatomy and palpation) for improving outcomes. This retrospective study aims to review our clinical outcomes of treating essential tremor (ET) and Parkinson’s disease (PD) tremor using either clinical- or kinematic-based injection pattern determination methods. Methods: 68 ET and 45 PD patients received at least one injection for their upper-limb tremor (unilateral or bilateral) in the last 7 years. Demographics of patients and BoNT-A injections were collected. A Mann–Whitney U statistical test was used to compare outcome measures between ET and PD cohorts. Results: Mean age (72 ± 9 years), number of injections (5), years receiving therapy (~2 years), clinic- (~57%) or kinematic-based patterns, and self-paying (52%) were similar between both cohorts. BoNT-A as a monotherapy in both upper limbs was received in more ET than PD patients. Double reconstitution of Xeomin^®^ in the wrist flexors/extensors, supinator, biceps, and triceps were most injected. Discontinuation due to no benefit/weakness was not dependent on the injection pattern determination approach. Conclusions: Kinematic-based BoNT-A injections produced similar treatment outcomes to injections based on the clinical expertise of the expert injector. This suggests that kinematics could be used by a non-expert to attain equivalent efficacy potentially improving access to this treatment.

## 1. Introduction

Upper-limb tremor often interferes with daily activities, impacts social lifestyles, and reduces the quality of life in essential tremor (ET) and Parkinson’s disease (PD) patients [1,2,3]. Oral medications, such as beta blockers for ET patients and dopaminergic replacement therapy (DRT) for PD patients, provide suboptimal benefit [4,5,6,7,8]. Approximately 30% of PD patients are refractory to DRT, and 60% of ET patients discontinue oral medications due to failure to alleviate tremor and side effects [4,6,9,10,11]. Botulinum toxin type A (BoNT-A) as a targeted tremor therapy holds promise when personalizing injection patterns (selection of muscle and dose) to the characteristics of an individual’s tremor [9,12,13,14,15]. Objective techniques to personalize injection pattern determination include using kinematic technology that measures the severity (selection of dose) and the contribution of tremor in all the upper limb joints (selection of muscles) or using the Yale method that involves measuring rhythmic burst potentials of electromyographic (EMG) activity, and the dose is based on the activity and size of the muscle [9,12,13,14,15]. Personalized injection approaches result in the improvement of tremor severity with lower rates of muscle weakness, as muscles contributing to tremor are injected with dosages that are based on the severity of the tremulous movements [9,12,13,14,15]. Previously published trials using fixed dosing/muscle patterns have shown limited functional improvements and marked/severe muscle weakness owing to the lack of personalization of injection patterns [16,17].

Literature describing how to initiate therapy using these technology-based methods (the Yale method [14,15] or kinematics [18]) or using a pragmatic approach to dose and to select muscles are available [19]. Technology-based dosing methods provide objective tremor analysis for all severities of whole upper-limb tremor, thereby offering a standardized and easier approach for BoNT-A pattern decision making [9]. Using the knowledge of surface anatomy and palpation is crucial for localization of some forearm muscles (e.g., wrist extensors and flexors) during BoNT-A administration, but this approach for dosing is highly subjective and dependent on the expertise of the injector and is not feasible for treating whole upper-limb tremor [9]. However, it is unclear in real-world practices whether a technology-guided approach can compare with expert clinical assessments (including surface anatomy and palpation) in the improved outcomes [20]. Thus, our real-world longitudinal experience of BoNT-A therapy for upper-limb ET and PD tremor is reported in this retrospective study. The aim is to review our clinical experience for treating upper-limb ET and PD tremor using BoNT-A. Injection patterns were determined using either clinical/visual assessment or technology-guided (kinematics) methods in both ET and PD cohorts. A cohort of ET and PD patients who completed our group’s pilot studies [12,13,21] and continued receiving BoNT-A therapy in the clinic were included in this retrospective analysis. The dosing pattern ultimately utilized in the clinic was optimized in these pilot studies [12,13,21]. The participants of these pilot studies were treated using BoNT-A dosing patterns determined by the clinical- and kinematic-based method [21] or solely by the kinematic-based dosing method (no clinical assessment) [12,13].

## 2. Results

### 2.1. Study Population

The demographics of the ET and PD patient cohorts are summarized in Table 1. Mean age and sex ratio were similar between the ET and PD patient cohorts. BoNT-A injections as a monotherapy were performed in significantly more (U = 907.5, *p* < 0.001) ET patients compared to PD patients. Furthermore, bilateral upper-limb injections were administered in significantly more (U = 935.5, *p* < 0.001) ET patients compared to PD patients.

### 2.2. BoNT-A Therapy

BoNT-A injection demographics for ET and PD cohorts are summarized in Table 2. The majority (86%) of the ET and PD patients received Xeomin^®^ compared to Botox^®^ BoNT-A formulations. The ratio of clinic-based to kinematic-based injection pattern determination was not significantly different (*p* = 0.264) between the ET and the PD cohorts. 

All patients except those optimized in the pilot studies initially received BoNT-A treatments using a fixed 3-month cycle. This fixed-interval injection cycle continued for 18 (26%) ET and 14 (31%) PD patients. However, a flexible injection cycle interval was introduced if a patient-perceived tremor reduction was maintained at the 3-month time-point, or the patient was self-paying for the drug. Thus, 42 (62%) ET and 25 (56%) PD patients followed a flexible injection timeline (Figure 1). The total number of injections (~5 injection cycles), the number of years (~2 years) of BoNT-A therapy, and the average interval per cycle (~5 months) were similar between ET and PD cohorts. Clinic-based or kinematic-based methods of injection determination did not influence (*p* > 0.05) the injection interval cycle or the number of injection cycles/number of years receiving therapy for both cohorts.

### 2.3. BoNT-A Dosing

The most frequently injected muscles reported in both cohorts were the flexor carpi radialis (FCR), flexor carpi ulnaris (FCU), extensor carpi radialis (ECR), extensor carpi ulnaris (ECU), supinator, biceps, and triceps. Wrist, elbow, and shoulder muscle dosages ranged between 5–20 U, 10–40 U, and 5–50 U, respectively, in both cohorts. The total mean dose per limb was significantly lower (U = 2202.5, *p* = 0.049) for ET patients (119.2 ± 55.4 U) compared to PD patients (138.2 ± 51.3 U). Mean total dose per limb was not significantly different depending on whether injection patterns were determined using clinic- or kinematic-based methods for both cohorts (ET: *p* = 0.798, PD: *p* = 0.680) (Table 3).

During dose optimization, 20 (29%) ET patients required an increase in dose (n = 7 clinic, 10 kinematic, 3 study), whereas 11 (16%) required a dose decrease (n = 6 clinic, 4 kinematic, 1 study). In the PD cohort, 13 (29%) patients required a dose increase (n = 11 clinic, 1 kinematic, 1 study), and 3 (7%) required a dose decrease (n = 3 kinematic).

### 2.4. BoNT-A Therapy Discontinuation

Between June 2015 and January 2022, eight (12%) ET and six (13%) PD patients received one injection and discontinued due to either no benefit or the cost of the drug, and twenty-three (34%) ET and seventeen (38%) PD patients discontinued due to side-effects/no tremor benefit, cost of drug, or were lost to follow-up. For the 23 out of the total 68 ET patients who discontinued following at least two cycles, 4 (6%) experienced muscle weakness, 3 (4%) could not afford injection costs, 6 (9%) perceived no benefit, 6 (9%) were lost to follow-up, 2 (3%) died of unrelated causes, and 2 (3%) had changes to their health unrelated to BoNT-A. For the 17 out of the 45 PD patients who discontinued, 5 (11%) experienced muscle weakness, 6 (13%) perceived no benefit, 4 (9%) were lost to follow-up, and 2 (4%) chose to continue solely using their PD medication. The reasons for discontinuation of BoNT-A therapy by method of injection pattern determination per cohort are summarized in Figure 2. There were no significant differences in the number of patients who received clinic- or kinematic-based injections and withdrew due to weakness/no benefit. Thus, 37 (54%) ET and 22 (49%) PD patients continued to receive serial BoNT-A therapy.

## 3. Discussion

This study reports real-world experience for treating upper-limb ET and PD tremor using either clinic-based by an expert injector versus kinematic-based BoNT-A injections in 113 patients over a total period of 7 years. The total dose per upper limb, number of patients on a flexible interval injection cycle, and the number of years receiving therapy were similar between patients receiving clinic- or kinematic-based BoNT-A injections for both cohorts. Total percentage of patients (18.5%) who discontinued therapy due to weakness/no benefit was similar between the two methods of injection pattern determination for both cohorts. In fact, our rate of discontinuation was close to the only available real-world, retrospective study of BoNT-A for refractory hand tremor (~15.4%) [20] and from our kinematic-tremor studies (ET cohort [12]: ~10%, PD cohort [13]: ~17%). This emphasizes BoNT-A injection patterns determined using kinematic tremor assessments can produce similar treatment outcomes as compared to the clinic-based assessment method performed by the expert injector. For both pattern determination approaches, the belly of the FCR, FCU, ECR, ECU, supinator, biceps, and triceps muscles were the most frequently injected using EMG injection guidance. BoNT-A dosages were selected based on tremor severity and either visually determined or measured using kinematics. The expert injector has more than 30 years of experience and has prior knowledge of using kinematics [12,13,18,21], possibly enhancing visual tremor assessments. Anatomical landmarking and injection technique can be taught, but practical expertise is crucial for selecting appropriate muscle and dosage patterns for efficacious tremor relief [9]. This makes an efficacious therapy unavailable to many due to the challenges in the assessment of tremor [22].

Kinematics is easy to learn and use and automatically creates an individual tremor profile by identifying the contribution of tremor in each degree of freedom and the severity of tremor per arm joint in under 15 minutes [23,24,25]. The severity of multi-joint tremor assessed using kinematics is correlated to the BoNT-A dose required to alleviate tremor [18]. Thus, our real-world experience suggests that a novice injector can use kinematics as an aid to develop the initial BoNT-A injection pattern. Furthermore, the number of patients who required changes to their injection pattern (dose increase or decrease) were similar between both methods of injection pattern determination for both cohorts. Thus, a novice injector could also optimize therapy based on patient feedback. Kinematic devices can cost up to CAD 10,000 and are not yet available for commercial sale. They are, however, available for research studies. They are fully approved for clinical use by the regulatory authorities in this context.

The potential for offering BoNT-A treatment can therefore be greatly enhanced in the community setting for tremor, which is very common. Localization of tremulous muscles using palpation and surface anatomy and treating wrist tremor with an average starting dose of 65 U, mostly in the FCR and FCU muscles [20], has been claimed to be more cost- and time-effective than the Yale technique and kinematic assessments [9]. Although similar tremor and functional improvements with low side-effect profiles were achieved, the kinematic dosing approach did not exclude severe tremor patients, and both the Yale and kinematic approaches treated tremor affecting proximal joints [9,20]. Thus, there is significant value to use technology to address unmet needs for treating tremor (e.g., suboptimal effect or cannot tolerate oral medications and limitations of surgical options) that both patients and physicians face [6,9,10].

A higher portion of ET patients received BoNT-A as a monotherapy compared to the PD patient cohort. This may be due to the adverse effect profile and the lack of benefit of oral medications experienced by ET patients and the availability and requirement of levodopa to treat other cardinal PD symptoms. It is interesting to note that dosages required for ET patients were lower than PD patients, and the dosage differences may be due to a lower severity of tremor observed in ET. A flexible injection cycle, an average of 5 months, and a double concentration reconstitution were preferred for both ET and PD cohorts (Table 2). These factors may aid in minimizing muscle weakness side effects while patients still perceive tremor relief [12,13]. A longer injection interval may also be utilized by some patients—although this exact number was not documented—to reduce cost, as approximately half of the patients must cover the cost of the drug. Furthermore, Xeomin^®^ BoNT-A formulation was used preferentially for both cohorts. The preference of using Xeomin^®^ compared to Botox^®^ is in part due to our group’s extensive experience of using Xeomin^®^ for tremor, as reported in the pilot studies [12,13,21], and due to the slightly lower overall cost of the drug in Canada for a variety of reasons.

The strength of this study is its practical, longitudinal design reporting BoNT-A therapy for tremor. However, there are limitations due to its retrospective nature. No specific clinical ratings of the change in tremor severity following each injection were documented. Prospective studies comparing the efficacy, safety, and dosages between BoNT-A injection patterns determined clinically or kinematically may help confirm our real-world results. However, a substantial number of patients continued to receive injections, which may suggest efficacy in the real-world settings, as has already been demonstrated in published studies [12,13,21]. Although no placebo-control was used, we do not think a placebo or bias response would account for patients returning for therapy, as most of the patients self-pay for the drug, travel long distances, and notice wearing off between doses, thus choosing flexible interval cycles. Furthermore, all patients who received BoNT-A were selected for BoNT-A due to either poor response or undesirable side effects of anti-tremor medications.

## 4. Conclusions

This retrospective study suggests BoNT-A therapy may provide beneficial, longitudinal treatment outcomes for upper-limb ET and PD tremor. Both clinic-based tremor assessment by an expert injector and kinematic-based injection patterns, which can be performed by a technician, produce similar treatment management outcomes. As all preliminary and controlled trial data and results have indicated a positive effect on tremor, clinical trials have begun for the use of both Botox^®^ and Xeomin^®^ in multiple centers. Recruitment is ongoing. One can hope that, if positive, injection of BoNT-A for tremor in general will be an approved indication soon.

## 5. Materials and Methods

### 5.1. Study Design

At the London Movement Disorders Centre in London, Ontario, Canada, 189 patients were treated with BoNT-A injections in the upper limb between June 2015 and January 2022. Patients diagnosed with ET or PD and receiving at least one BoNT-A injection for their upper-limb tremor (either unilateral or bilateral) were included for this retrospective analysis, approved by the Western University Health Sciences Research Ethics Board (# 120565) on 1 March 2022. Exclusion criteria were upper-limb tremor patients with other movement disorder indications (e.g., cerebral palsy, chorea, corticobasal syndrome, dystonia, generalized spasticity, multiple sclerosis, myoclonus, and progressive supranuclear palsy). Thus, from the 189 patient records, a total of 113 patients were included in this study (68 ET and 45 PD patients).

### 5.2. Outcome Measures

For the eligible 113 ET and PD patients, demographic information such as age, gender, optimized on anti-tremor medication (ET anti-tremor medications included: primidone, pramipexole, metoprolol, topiramate, gabapentin, and nadolol; PD anti-tremor medications included: levodopa, nadolol, gabapentin, selegiline, ropinirole, and parsitan) or received BoNT-A as a monotherapy, and the number of patients with unilateral or bilateral upper-limb tremor treated with BoNT-A injections was collected. Outcomes related to BoNT-A therapy included: BoNT-A formulation (Botox^®^, Allergan plc, Dublin, Ireland or Xeomin^®^, Merz Pharmaceuticals GmbH, Frankfurt am Main, Germany), BoNT-A reconstitution (single (1:1) or double (1:2) concentration), the method of BoNT-A injection pattern determination (clinical/visual assessment (“clinic-based”), a baseline kinematic tremor assessment using the previously published methodology (“kinematic-based”) [12,13,18], or a patient transferred to the clinic after completing the pilot tremor study [12,13,21], which utilized kinematic-based and optimized injection patterns (“optimized in pilot study”), total number of BoNT-A injection treatments, injection interval (in months), the number of years receiving BoNT-A, changes to the injection pattern (total dose increase, decrease or no change), the number of patients who discontinued BoNT-A therapy and the reason, and the number of patients required to self-pay for BoNT-A.

### 5.3. BoNT-A Injections

All injections were administered using needle-EMG guidance (The Myoguide System, Intronix Technologies Corporation, Bolton, ON, Canada) into the belly of the muscle. Upper-limb muscles injected were the flexor carpi radialis (FCR), flexor carpi ulnaris (FCU), extensor carpi radialis (ECR), extensor carpi ulnaris (ECU), pronator teres (PT), pronator quadratus (PQ), supinator, brachioradialis (BRD), biceps, triceps, pectoralis major (Pec. M), teres major (TM), supraspinatus, and deltoid. Total BoNT-A dose (BoNT-A units) per upper limb per patient, BoNT-A dose per muscle per patient, and the frequency of muscles injected were collected for both cohorts.

### 5.4. Statistical Analysis

A non-parametric, independent sample Mann–Whitney U statistical test was used to compare age, sex, number of patients receiving BoNT-A as a monotherapy, number of upper limbs treated per patient, method of injection pattern determination, mean number of injection visits, mean number of years receiving injections, injection interval duration (number of months per cycle), and mean total dose per upper limb between ET and PD cohorts and between the methods of injection pattern determination within each cohort (SPSS, version 21, IBM, Armonk, NY, USA, 2012).

## Figures and Tables

**Figure 1 toxins-14-00557-f001:**
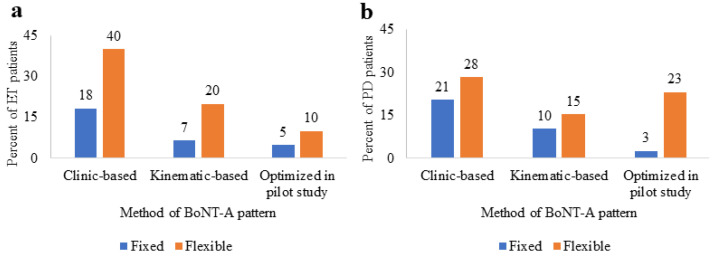
Percent of ET (**a**) and PD (**b**) patients who received BoNT-A using a fixed (3-month) or flexible injection cycle timeline depending on the method of BoNT-A injection pattern determination (clinic-based, kinematic-based, or kinematic-based and optimized in the pilot study). Percent data values are included for a total number of 60 ET and 39 PD patients.

**Figure 2 toxins-14-00557-f002:**
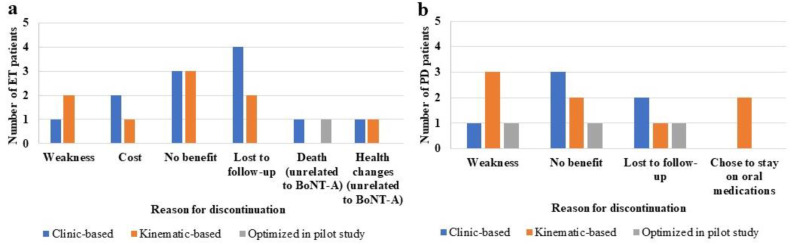
Number of ET (**a**) and PD (**b**) patients who discontinued BoNT-A therapy following at least 2 cycles separated by the method of BoNT-A injection pattern determination (clinic-based, kinematic-based, and kinematic-based and optimized from the pilot study).

**Table 1 toxins-14-00557-t001:** Demographics of the ET and PD patient cohorts.

	ET	PD
Number of patients enrolled (n)	68	45
Men:Women ratio (*n*)	36:32	28:17
Age (years) ^a^	72 ± 9.5 (min: 46, max: 94)	72 ± 9.2 (min: 50, max: 89)
Medication history ^b,c^		
Monotherapy BoNT-A	33 (50%)	7 (16%)
BoNT-A as an adjunct		
therapy (optimized on anti-tremor mediations)	33 (50%)	37 (82%)
Unilateral vs. bilateral tremor ^b^		
Left arm	7 (10%)	11 (24%)
Right arm	24 (35%)	27 (60%)
Both arms	37 (54%)	7 (16%)

^a^ Average, standard deviation, and min/max range per cohort. ^b^ Total number and percent of patients per cohort. ^c^ Data unavailable for 2 ET patients and 1 PD patient. Abbreviations: ET, essential tremor; max, maximum; min, minimum; *n*, number of patients; PD, Parkinson’s disease.

**Table 2 toxins-14-00557-t002:** BoNT-A therapy demographics for ET and PD upper-limb tremor.

	ET	PD
Botox^®^ to Xeomin^®^ formulation ratio (*n*) ^a^	8:49	6:35
1:1 to 1:2 BoNT-A reconstitution ratio (*n*) ^a^	16:41	10:31
Method of BoNT-A pattern ^b^		
Clinic-based	42 (62%)	23 (51%)
Kinematic-based	17 (25%)	10 (22%)
Kinematic-based and optimized in pilot study [12,13,21] ^c^	9 (13%)	12 (27%)
Number of injections ^a,d^	5.5 ± 4.2 (min: 1, max: 18)	5.3 ± 4.1 (min: 1, max: 15)
Number of years receiving BoNT-A ^a,c^	2.1 ± 1.1 (min: 1, max: 4)	1.9 ± 1.0 (min: 1, max: 4)
Injection interval cycle (months) ^a,c^	4.7 ± 1.6 (min: 2.4, max: 12)	4.2 ± 1.5 (min: 2.4, max: 8)
Self-paying (no health insurance coverage) ^b^	26 (46%)	24 (58%)

^a^ Data unavailable for 11 ET and 4 PD patients. ^b^ Total number and percent of patients per cohort. ^c^ A total of 21 patients who participated and completed the cited studies continued to receive BoNT-A therapy in the clinic. ^d^ Average, standard deviation, and min/max range per cohort. Abbreviations: ET, essential tremor; max, maximum; min, minimum; *n*, number of patients; PD, Parkinson’s disease.

**Table 3 toxins-14-00557-t003:** Mean BoNT-A dose summarized by the method of injection pattern determination in ET.

Mean Total Dose (U) Per Upper Limb	ET	PD
Clinic-based	118.2 ± 59.8 (min = 10, max = 245)	140.9 ± 57.0 (min = 15, max = 230)
Kinematic-based	136.9 ± 50.0 (min = 45, max = 250)	137.0 ± 51.0 (min = 25, max = 220)
Kinematic-based, optimized in pilot study	96.3 ± 34.3 (min = 45, max = 155)	134.5 ± 43.8 (min = 70, max = 245)

Abbreviations: ET, essential tremor; PD, Parkinson’s disease; U, Botox^®^/Xeomin^®^ units.

## Data Availability

Data are contained within the article.

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
