# Peer review of "Real-World Longitudinal Experience of Botulinum Toxin Therapy for Parkinson and Essential Tremor"

_toxins, 2022, doi:10.3390/toxins14080557_

Round 1
Reviewer 1 Report
In this manuscript, the authors compared the outcomes of the BoNT-A injection for Parkinson’s Disease and Essential Tremor Patients using either clinical/visual assessment by an expert injector or a kinematic-based injection.
This manuscript is interesting; however, this manuscript needs substantial improvements and corrections before publishing may be possible.
General points:
Please add a list of abbreviations before References section to your manuscript.
Special points:
Please change and correct the title of your manuscript: the title does not correspond with your study aim and your results.
Importantly, this manuscript should be improved, i. e., by substantial references in the field.
Abstract
Lines 6-7: please write out: ET and PD.
Introduction
Lines 26-27: please add multiple references at the end of this sentence.
Lines 27-29: please add more references at the end of this sentence.
Lines 29-31: please add more references at the end of this sentence.
Lines 31-38: please describe very exactly all these studies.
Results
Table 2: what did you mean with References 4, 5 and 13? Did you involve in this table not only your patients? Did you involve also patients from the literature?
Lines 83-83: please write out: FCR, FCU, ECR, ECU.
Figure 2: please improve the Figure: make bigger and in better quality.
Discussion
Lines 131-132: please describe all these landmarking and injection points, and add the appropriate references.
Lines 134-135: please add multiple references at the end of this sentence.
Lines 136-138: please add multiple references at the end of this sentence.
Conclusions
Please also add a Future perspectives section.
Materials and Methods
Lines 182-187: please add also the exact date for the permission of your study.
Line 209: please add multiple references at the end of this sentence.
Lines 209-215: please add to your manuscript as a Figure the injection points of BoNT-A in ET and PD patients.
References
Please do your References list according to “Toxins”.
Author Response
Please add a list of abbreviations before References section to your manuscript.
- A list of abbreviations have been added to page 8.
Special points:
Please change and correct the title of your manuscript: the title does not correspond with your study aim and your results.
- The title has been changed to reflect the manuscript.
- Removed “outcomes” and replaced with “experience”
Importantly, this manuscript should be improved, i. e., by substantial references in the field.
- References have been added where reviewer has stated below (a total of 12 references have been added)
Abstract
Lines 6-7: please write out: ET and PD.
- The abbreviations have been written out
Introduction
Lines 26-27: please add multiple references at the end of this sentence.
Lines 27-29: please add more references at the end of this sentence.
Lines 29-31: please add more references at the end of this sentence.
- References have been added where asked. References 1-3, 4-8 and 12-15 have been added to the introduction.
Lines 31-38: please describe very exactly all these studies.
- The details of these studies have been clarified: Botulinum toxin type A (BoNT-A) as a targeted tremor therapy holds promise when personalizing injection patterns (selection of muscle and dose) to the characteristics of an individual’s tremor [9,12-15]. Objective techniques to personalize injection pattern determination include using kinematic technology to measure the severity (selection of dose) and the contribution of tremor in all upper limb joints (muscles) or using the Yale method that involves measuring rhythmic burst potentials of electromyographic (EMG) activity and the dose is based on the activity and size of the muscle [9,12-15].
Results
Table 2: what did you mean with References 4, 5 and 13? Did you involve in this table not only your patients? Did you involve also patients from the literature?
- We have clarified that patients from these references are in our clinic. Last sentence of the introduction: “A cohort of ET and PD patients who completed our group’s pilot studies [12,13,21] and continued receiving BoNT-A therapy in the clinic were included in this retrospective analysis.” And also clarified in the legend of Table 2: “A total of 21 patients who participated and completed the cited studies continued to receive BoNT-A therapy in the clinic”
Lines 83-83: please write out: FCR, FCU, ECR, ECU.
- These abbreviations have been written out in full
Figure 2: please improve the Figure: make bigger and in better quality.
- The font of the figure has been increased, the figure has been enlarged and with better quality (600 dpi).
Discussion
Lines 131-132: please describe all these landmarking and injection points, and add the appropriate references.
- For both pattern determination approaches, the belly of the FCR, FCU, ECR, ECU, supinator, biceps, and triceps muscles were the most frequently injected using EMG injection guidance. BoNT-A dosages were selected based on tremor severity, either visually determined or measured using kinematics.
Lines 134-135: please add multiple references at the end of this sentence.
Lines 136-138: please add multiple references at the end of this sentence.
- References have been added: (#12,13,18,21) and #9 and #22.
Conclusions
Please also add a Future perspectives section.
RESPONSE: As all preliminary and controlled trial data and results have indicated a positive effect on tremor, clinical trials have begun for the use of both Botox and Xeomin in multiple centres. Recruitment is ongoing. One can hope that if positive, injection of botulinum toxin for tremor in general will be an approved indication soon.
Materials and Methods
Lines 182-187: please add also the exact date for the permission of your study.
- The approval date of the REB for this study has been added: March 1, 2022.
Line 209: please add multiple references at the end of this sentence.
- References have been added: 12, 13, 18 and 21.
Lines 209-215: please add to your manuscript as a Figure the injection points of BoNT-A in ET and PD patients.
- We have clarified that the belly of the muscles listed were targeted using needle EMG guidance on page 7. And also in the discussion: “For both pattern determination approaches, the belly of the FCR, FCU, ECR, ECU, supinator, biceps, and triceps muscles were the most frequently injected using EMG injection guidance.”
References
Please do your References list according to “Toxins”
- The reference list has been amended according to the journal’s guidelines on page 8.
Reviewer 2 Report
Abstract:
Line 5: There is real world experience about BoNTA-A use in tremors. Niemann and Jankovic, 2018 in their article “Botulinum Toxin for the Treatment of Hand Tremor”, which was a retrospective review. Please make your statement more accurate.
Introduction:
Line 39: “it is unclear in real-world practices whether a technology-guided approach can compare with expert clinical assessments (including surface anatomy and palpation) in the improved outcomes “. This statement highlights the strength of your article, it would be best if you added this to the background of your abstract.
Results: Line 86: It's curious that the dosages required for your ET patients are lower than PD. Do you have any thoughts on why this may be? If so, please state it there.
Line 95: Did any of your patients with anatomic guidance BoNT-A receive injections under EMG or ultrasound guidance?
Discussion:
Line 143: A novice injector, who has completed a movement disorders fellowship, would likely be more familiar with anatomic guidance for BoNT-A injections, rather than kinematic guidance. Kinematic guidance is not prevalent widely.
Are the kinematic devices inexpensive, can someone starting practice easily invest in such a device? Can you please make a comment based on your experience?
Please try to include a series of images with steps on the usage of kinematic guidance or a video depicting the same. Or, make references to articles that already have videos that are instructive.
Author Response
Abstract:
Line 5: There is real world experience about BoNTA-A use in tremors. Niemann and Jankovic, 2018 in their article “Botulinum Toxin for the Treatment of Hand Tremor”, which was a retrospective review. Please make your statement more accurate.
- We have amended this statement to say: However, it is unclear in real world practices whether a technology-guided approach can compare with expert clinical assessments (including surface anatomy and palpation) for improving outcomes.
Introduction:
Line 39: “it is unclear in real-world practices whether a technology-guided approach can compare with expert clinical assessments (including surface anatomy and palpation) in the improved outcomes “. This statement highlights the strength of your article, it would be best if you added this to the background of your abstract.
- We have added this statement to the abstract
Results: Line 86: It's curious that the dosages required for your ET patients are lower than PD. Do you have any thoughts on why this may be? If so, please state it there.
- We have added this to the discussion on page 6: It is interesting to note that dosages required for ET patients were lower than PD patients and the dosage differences may be due to a lower severity of tremor observed in ET.
Line 95: Did any of your patients with anatomic guidance BoNT-A receive injections under EMG or ultrasound guidance?
- We have clarified the manuscript that all injections were conducted under EMG guidance.In the methods on page7: “ All injections were administered using needle-electromyography guidance into the belly of the muscle” and in the discussion on page5: “ For both pattern determination approaches, the belly of the FCR, FCU, ECR, ECU, supinator, biceps, and triceps muscles were the most frequently injected using EMG injection guidance.”
Discussion:
Line 143: A novice injector, who has completed a movement disorders fellowship, would likely be more familiar with anatomic guidance for BoNT-A injections, rather than kinematic guidance. Kinematic guidance is not prevalent widely.
- We have clarified the ending of the first discussion paragraph to state this: “BoNT-A dosages were selected based on tremor severity, either visually determined or measured using kinematics. The expert injector has more than 30 years of experience and has prior knowledge of using kinematics [12,13,18,21] possibly enhancing visual tremor assessments. Anatomical landmarking and injection technique can be taught but practical expertise is crucial for selecting appropriate muscle and dosage patterns for efficacious tremor relief [9]. This makes an efficacious therapy unavailable to many due to the challenges in the assessment of tremor [22].”
Are the kinematic devices inexpensive, can someone starting practice easily invest in such a device? Can you please make a comment based on your experience?
- We have included a statement in the discussion on page 5-6: Kinematic devices can cost up to 10,000 Canadian dollars and are not as yet available for commercial sale. They are however available for research studies. They are fully approved for clinical use by the regulatory authorities in this context.
Please try to include a series of images with steps on the usage of kinematic guidance or a video depicting the same. Or, make references to articles that already have videos that are instructive.
- We have added references to how kinematic guidance is done and dosing is calculated: references 18, 23-25.
Reviewer 3 Report
Thank you for this summary of your data and status.
The results are undoubtedly interesting and worthy of publication.
However, one would wish for a somewhat more critical discussion.
Critical literature should also be acknowledged.
Author Response
Thank you for this summary of your data and status.
The results are undoubtedly interesting and worthy of publication.
However, one would wish for a somewhat more critical discussion.
- We have added to the discussion first paragraph: “For both pattern determination approaches, the belly of the FCR, FCU, ECR, ECU, supinator, biceps, and triceps muscles were the most frequently injected using EMG injection guidance. BoNT-A dosages were selected based on tremor severity, either visually determined or measured using kinematics. The expert injector has more than 30 years of experience and has prior knowledge of using kinematics [12,13,18,21] possibly enhancing visual tremor assessments. Anatomical landmarking and injection technique can be taught but practical expertise is crucial for selecting appropriate muscle and dosage patterns for efficacious tremor relief [9]. This makes an efficacious therapy unavailable to many due to the challenges in the assessment of tremor [22].”
- As well we have added to the end of the second discussion paragraph: “Kinematic devices can cost up to 10,000 Canadian dollars and are not as yet available for commercial sale. They are however available for research studies. They are fully approved for clinical use by the regulatory authorities in this context.”
- WE have added a 3rd paragraph stating this: “The potential for offering BoNT-A treatment can therefore be greatly enhanced in the community setting for tremor which is very common. Localization of tremulous muscles using palpation and surface anatomy and treating wrist tremor with an aver-age starting dose of 65 U, mostly in the FCR and FCU muscles [20], has been claimed to be more cost and time effective than the Yale technique and kinematic assessments [9]. Although, similar tremor and functional improvements with low side effect profiles were achieved, the kinematic dosing approach did not exclude severe tremor patients and both the Yale and kinematic approaches treated tremor affecting proximal joints [9,20]. Thus, there is significant value to use technology to address unmet needs for treating tremor (e.g. suboptimal effect or cannot tolerate oral medications, and limita-tions of surgical options) that both patients and physicians face [6,9,26].”
- And a short statement to the 4th paragraph: “It is interesting to note that dosages required for ET patients were lower than PD patients and the dosage differences may be due to a lower severity of tremor observed in ET.”
Critical literature should also be acknowledged.
- We have added references where literature must be acknowledged as this was identified by reviewer #1. In total we have 25 references, an addition of 12 references in this revision.
Round 2
Reviewer 1 Report
Dear authors, thank you for all corrections. Unfortunately, this manuscript needs some additional corrections before publishing may be possible. Introduction Lines 45-46: please describe very exactly all these studies. Lines 53-55: please describe very exactly all these studies.Author Response
Introduction
Lines 45-46: please describe very exactly all these studies.
- We have added this following the beginning of the 2nd introduction paragraph: “Literature describing how to initiate therapy using these technology-based methods (the Yale method [14-15] or kinematics [18]) or using a pragmatic approach to dose and to select muscles are available [19]. Technology-based dosing methods provide objective tremor analysis for all severities of whole upper limb tremor thereby offering a standardized and easier approach for BoNT-A pattern decision making [9]. Using the knowledge of surface anatomy and palpation is crucial for localization of some forearm muscles (e.g. wrist extensors and flexors) during BoNT-A administration, but this approach for dosing is highly subjective and dependent on the expertise of the injector and is not feasible for treating whole upper limb tremor [9].”
Lines 53-55: please describe very exactly all these studies.
- We have added this to the end of the 2nd introduction paragraph: “A cohort of ET and PD patients who completed our group’s pilot studies [12,13,21] and continued receiving BoNT-A therapy in the clinic were included in this retrospective analysis. The dosing pattern ultimately utilized in the clinic were optimized in these pilot studies [12, 13, 21]. The participants of these pilot studies were treated using BoNT-A dosing patterns determined by the clinical and kinematic-based method [21] or solely by the kinematic-based dosing method (no clinical assessment) [12,13].”
- In addition, minor changes were done in the introduction, first sentence of the discussion and changed botulinum toxin to BoNT-A in the last sentence of the conclusion.